# The Utilization of Early Outpatient Care for Infants Following NICU Discharge among a National Sample

**DOI:** 10.3390/children11050550

**Published:** 2024-05-04

**Authors:** Janine P. Bernardo, Lisa Yanek, Pamela Donohue

**Affiliations:** 1Division of Neonatology, Department of Pediatrics, Massachusetts General Hospital, Boston, MA 02114, USA; 2Department of Medicine, Johns Hopkins University School of Medicine, Baltimore, MD 21287, USA; lryanek@jhmi.edu; 3Division of Neonatology, Department of Pediatrics, Johns Hopkins University School of Medicine, Baltimore, MD 21287, USA; pdonohue@jhmi.edu

**Keywords:** neonatal intensive care unit, outpatient, follow-up, health services, high risk, readmission, national

## Abstract

Outpatient care following discharge from a neonatal intensive care unit (NICU) is critical for streamlined transfer of care. Yet, information is lacking about the characteristics of early outpatient care. The objective of this secondary data analysis is to describe outpatient encounters (OPEs) within the first three months following the discharge of commercially insured infants admitted to NICUs in the MarketScan Research Database nationally from 2015 to 2017. Data were analyzed using descriptive statistics and logistic regression. A total of 22,214 NICU survivors were included, of whom half had an OPE within two days following discharge (quartiles 1, 3) and 90% within five days. The median number of OPEs in the first three months was five (quartiles 4, 7). A majority of first physician visits were with pediatricians (81.5%). A minority of infants with chronic conditions saw subspecialists. Term infants with delayed care had a lower risk of readmission. Spending was higher for preterm infants and those with chronic conditions. We conclude that most patients are seen shortly after discharge and by pediatricians; however, there is room for improvement. Frequent encounters and spending afflict high-risk groups with chronic conditions. Future work should examine the associations of early outpatient care with social determinants of health and other outcomes such as immunizations.

## 1. Introduction

Early outpatient care for infants discharged from a neonatal intensive care unit (NICU) is essential for addressing ongoing medical needs, supporting families, and establishing a medical home outside of the hospital [1,2]. Data on the outpatient service utilization of infants discharged from NICUs, including encounters with primary care providers, subspecialties, and other services, however, are limited, and research on early healthcare utilization focuses largely on readmission [3,4,5,6,7,8]. The evidence that exists on outpatient services and prescription drug use shows that preterm neonates are high utilizers of these services [9,10,11,12,13,14]. Previous work, however, lacks important descriptors of outpatient follow-up immediately following NICU discharge, instead exploring service use months to years later [15]. This work also focuses on the smallest, most premature babies, who represent a high-risk, albeit small, proportion of infants in most NICUs. Term neonates, who may represent nearly half of NICU admissions, are noticeably absent from analyses [16,17].

Establishing outpatient services shortly after discharge is a critical and important step towards transitioning the care of high-risk neonates. The American Academy of Pediatrics recommends that, prior to discharge, appropriate follow-up be established with primary care physicians, subspecialties, and skilled home nursing care, and that other necessary coordination occurs for needed medication and home equipment, without specifying recommendations based on gestational age or diagnoses [18]. The initial appointment for follow-up should be within 1–2 days following discharge in order to facilitate continuity of care [2]. However, there are limited data on actual compliance with these recommendations or knowledge of other aspects of early outpatient services. Despite the body of literature focusing on readmissions of NICU infants, it is unknown whether the timing of outpatient encounters is associated with readmission risk. The cost of outpatient services is also underexplored [3,15].

Questions therefore exist regarding the outpatient care of preterm and term NICU infants soon after discharge. In this work, our objectives were to describe aspects of early outpatient utilization within three months of NICU discharge among a national sample, including the timing of the initial encounter, the frequency of encounters, the location of services, the association of timing with early readmission, and spending on outpatient encounters. Such knowledge can inform safe and well-coordinated NICU discharge planning, identify neonates at risk for delayed follow-up care, and may prevent hospital readmission, thereby lowering healthcare spending.

## 2. Materials and Methods

### 2.1. Study Design and Population 

The study population included infants with commercial insurance identified through data extracted from IBM MarketScan Research Databases^®^. Descriptors of clinical care, medical resource utilization, and expenditures are linked by unique person-level identifiers across inpatient and outpatient encounters. Dates, location of service (both geographic and clinical location such as emergency room, physician office, etc.), and type of provider are available. Data included approximately 350 payers and represented all 50 states. 

For this study, outpatient data were linked to a previously published inpatient sample of infants billed under hospital revenue code 174, which indicates admission to newborn intensive care [8]. Patients were born from January 2015 to December 2017; outpatient data for the first three months after hospital discharge were extracted. Because the database represents insurance claims, only patients with the same insurer during the inpatient and outpatient study period could be included. Per IBM MarketScan data documentation, a small percentage of claims coded as outpatient services represent inpatient care; these data were identified and removed from the analytic file. Among all outpatient data, missingness did not exceed 1.5% for any variable.

The study used entirely de-identified data, was reviewed by the Institutional Review Board of Johns Hopkins Medicine, and was deemed exempt under Federal Regulations (45 CFR 46) use of secondary data.

### 2.2. Demographic and Clinical Characteristics of NICU Infants 

Methods for the extraction of inpatient NICU data have been previously published [8]. Demographic characteristics were collected for NICU infants including NICU length of stay (LOS) and readmission rates during the first three months after NICU discharge. LOS was defined as the total number of hospital days at referring hospitals and subsequent step-down units for those transferred prior to discharge. Gestational age (GA), categorized as preterm (<37 weeks) or full-term, and diagnostic categories were constructed using ICD-10 codes from inpatient NICU data previously described [8]. Specifically, they included the following: moderate and severe hypoxic–ischemic encephalopathy (HIE); grade III or IV intraventricular hemorrhage or periventricular leukomalacia (IVH/PVL); Bell stage 2 or 3 necrotizing enterocolitis or spontaneous intestinal perforation (NEC/SIP); congenital heart disease (CHD), which excluded infants with patent ductus arteriosus (PDA) only. Four chronic condition variables were created based on NICU diagnoses: chronic lung disease/bronchopulmonary dysplasia (CLD/BPD), CHD (as described); NEC/SIP; and brain injury (BI) defined as HIE or IVH/PVL. Inclusion in these variables is overlapping; for instance, an infant may be included in the CLD/BPD variable and the BI variable. There was no more than 30% overlap for any combination of chronic conditions. For those discharged home with equipment, the equipment included a gastrostomy tube (GT), tracheostomy (trach), oxygen, oxygen with ventilation, or ventriculoperitoneal shunt (VPS).

### 2.3. Descriptors of Outpatient Encounters

Outpatient encounters (OPEs) were defined as any encounter with a physician, laboratory, imaging or pharmacy services, physical therapists (or other therapists), supply center, hospice, or home healthcare (Appendix A). Data were also disaggregated into physician encounters (PEs) or non-physician encounters (NPEs). Preterm and term infants were described separately due to differing characteristics, diagnoses, and lengths of stay.

Among infants with a PE, physicians were grouped by specialty to examine which infants saw general pediatricians versus other types of physicians, including subspecialists (pediatric and non-pediatric), family practitioners, and others, coded only as an unspecified medical doctor or multispecialty physician group (Appendix A). Nurse Practitioners (NPs) and Physician Assistants (PAs) were included among pediatricians as they generally practice with physicians and represented less than 1% of the data, likely due to billing occurrences under physicians.

The timing of outpatient encounters following NICU discharge was determined by examining outpatient service dates compared to the date of NICU hospital discharge. For infants transferred to a higher or lower level of care, the NICU hospital discharge date represents the date of the final hospital discharge prior to going home. Additionally, we calculated the number of encounters and described the geographical region of outpatient services.

The time to first OPE (OPE1) was examined by quartiles. Patients in the highest quartile of time to OPE1 (only including encounters with a physician or mid-level provider) were defined as having delayed care (DC). These infants’ demographic and clinical characteristics were compared to all other infants in order to identify those children at risk for DC and to evaluate their risk for hospital readmission.

### 2.4. Spending on Outpatient Encounters 

Outpatient spending was reported as the total gross payment made by the insurers for the OPE. Negative values, of which there were few (<1%), were presumed to be errors or corrections for overcharges and were recoded as zero. Median spending was described for spending on the first outpatient encounter, including spending based on provider type, as well as spending on outpatient services during the first three months following discharge, also according to term/preterm and chronic conditions.

### 2.5. Statistical Approach

Categorical variables were described using frequencies and proportions, and continuous variables were described using medians and interquartile ranges. Chi-square tests were utilized to compare categorical variables and Mann–Whitney U tests to compare continuous variables. The risk of rehospitalization for infants with DC was calculated using simple logistic regression. All factors found to be statistically significant in the bivariate analysis were included in a multivariable regression model. Models were created for full-term and preterm infants due to differing characteristics among these groups. Diagnoses, including descriptors and clinical diagnoses, were only included in the models that affected at least 1% of infants. Model goodness of fit was examined with the Hosmer–Lemeshow test. Data were analyzed with IBM SPSS Statistics for Windows, Version 28.0. Armonk, NY, USA: IBM Corp.

## 3. Results

### 3.1. Population Description

The study population included 22,214 infants who survived until NICU discharge, had known GA (preterm/full-term), and had an OPE1 during the study period (Table 1). The majority of infants were preterm (64%) and male (57%). The median NICU LOS was 8 days but varied considerably if term versus preterm. The percentage of infants declined from 2015 to 2017. The most common NICU diagnoses were respiratory distress, observation or treatment of infection, and the presence of a congenital anomaly. The least common NICU diagnoses included HIE, NEC/SIP, and BPD/CLD. The prevalence of infants readmitted to the hospital by 7 days, 8–30 days, and three months after discharge were 3.1%, 2.5%, and 7.9%, respectively.

### 3.2. Descriptors of Outpatient Encounters in the First Three Months Following Discharge

The initial outpatient encounter had a median occurrence two (quartiles 1, 3) days after NICU discharge, with 90% being seen by five days (Table 2). The results were similar for the timing of the first physician encounter. Infants had a median of 5 (quartiles 4, 7) encounters within three months following NICU discharge, with a range from 1 to 90 total encounters. Infants with chronic NICU diagnoses of CLD, NEC/SIP, and BI had the highest number of outpatient encounters during the first three months following discharge. In comparisons between full-term and preterm infants, the number of encounters, the timing of encounters, and the distribution of the type of encounters, there were no clinical differences noted (Table 2). Patients with ninety encounters in the first three months, of whom there were five, were those with daily home health or hospice care. 

The most common type of outpatient encounter was with a physician. The percentage of patients who did not have encounters after OPE1 increased over time (Figure 1A). Among patients with an encounter, 73–79% saw a physician. Of those patients seen by a physician for the first visit, 81.5% were seen by a pediatrician (Figure 1B). Not surprisingly, during subsequent encounters, the percentage of patients seen by subspecialists increased, while those seen by pediatricians declined. A minority of infants with chronic diagnoses at the time of NICU discharge saw subspecialists within the first three months after discharge; 15.3% with CLD saw a pulmonologist, 19.7% with CHD saw a cardiologist, 10.3% with NEC/SIP saw a surgeon, and 4% with BI saw a neurologist, although a greater proportion of infants with HIE were seen by a neurologist (21.4%) than those with severe IVH/PVL (2.6%).

### 3.3. Delayed Care

To examine which patients were seen later by a physician or mid-level provider than other discharged infants, we compared patients in the highest quartile of time to OPE1 (delayed care, DC) with those seen earlier (Table 3). Fewer preterm and term infants with DC were readmitted (6.4% and 7.1%, respectively) by 3 months compared to those with timely care (7.5% and 9.3%). However, among those readmitted, the duration of readmission was significantly longer for those with DC than for others. When DC was defined as seven days or later, similar results were observed.

In a multiple regression model, DC was associated with reduced risk of readmission among term infants by three months (Table 4). Additionally, the presence of a congenital anomaly and being born in the Northcentral US was associated with readmission. Among preterm infants, NICU diagnoses of a congenital anomaly and NEC/SIP, as well as NICU LOS, were independently associated with readmission by three months, while SGA status was protective against readmission (Table 5).

### 3.4. Spending on Outpatient Encounters in the First Three Months Following NICU Discharge

The median spending on the first outpatient encounter for all providers and all discharged infants was USD 138 (Table 6A). The median spending for OPE1 with a pediatrician or subspecialists was USD 130 and USD 159, respectively. The median spending on home healthcare for the first visit was USD 172. Spending was higher during the first three months for preterm infants compared to full-term infants. Neonates with chronic diagnoses had 2–3 times the spending on outpatient encounters compared to all other infants (Table 6B).

## 4. Discussion

This study utilized a large, national database over three years to examine early outpatient service utilization for infants following NICU discharge. The results are encouraging; they suggest that at least 50% of infants are seen within two days following NICU discharge and 75% are seen within three days. NICU infants had a median of five encounters within the first three months following discharge, which is greater than the number of routine well-child visits suggested for newborns by the AAP [19]. Not surprisingly, infants with chronic conditions such as CLD and NEC/SIP had a higher number of visits than the general population. The majority of infants are seen by pediatricians, suggesting that pediatricians are carrying the greatest responsibility of early follow-up care for these vulnerable and often complex infants after discharge. While timely follow-up is desirable, it does not necessarily translate to the establishment of the medical home: comprehensive, family-centered, high-quality care that all infants deserve [20]. However, evidence that the majority of NICU infants are being seen by pediatricians soon after discharge is an important step towards acquiring this critical foundation of medical care. Further work on the establishment of the medical home within this population is needed. 

While many patients are seen in a timely manner, there are some that do not have encounters for weeks to even months following discharge, which may delay important medical care. Barriers to follow-up care, such as cost, transportation, and child care, could not be explored within this dataset but have been suggested as obstacles to follow-up, as well as other social and behavioral factors [21,22]. Such barriers remain important considerations to improve the timely establishment of follow-up care. 

We also found that subspecialist follow-up does not occur for many infants with chronic conditions within three months following discharge. Delayed pediatric subspecialty care may be due to the growing needs and complexity of surviving children and changes in the pediatric work force [23,24]. Regardless of the reasons, this places a greater responsibility of caring for medically complex infants on primary care providers. Further work can be conducted to explore the timing of subspecialty follow-up and to support infants and families who require subspecialty care. Importantly, the general pediatric work force who have assumed the early medical care of these children may be better assisted. A variety of mechanisms may better support pediatricians, including higher reimbursement rates, greater time with complex patients, investment in case managers to assist with care coordination, and more education on the primary care of NICU infants [25]. Recommendations exist on how to optimize the safe and streamlined transfer of care from the NICU to outpatient providers and include identifying a primary care provider, often a pediatrician, communicating pertinent medical information through written or oral means, and following up with families after discharge [1]. Attention to important aspects of NICU transition to the medical home, to the benefit of the patients and pediatricians, is critical, such as clear short- and long-term plans of care, arranged follow-up appointments, and warm handoffs for complex patients [1].

Studies among adult, pediatric, and newborn patients suggest the importance of early outpatient follow-up after hospitalization in reducing readmission [26,27,28,29]. While not the primary focus of this work, we nonetheless sought to explore an association between the timing of outpatient follow-up and readmission. Interestingly, in our study, delayed initial outpatient care was associated with a lower risk of readmission among term infants only. These results may reflect that a closer evaluation from a provider may inform vigilance from caregivers or that an evaluation by a provider is necessary prior to hospital admission. Examining this trend among patients with public and commercial insurance would be of interest. Previous work demonstrated that the characteristics of outpatient facilities influence the readmission rates of former NICU infants, suggesting that the quality of outpatient follow-up, perhaps more than the timing, plays an important role in readmission [22]. 

Our work adds to the discourse on substantial spending on NICU infants, impacting both healthcare systems and individual patients and families [30,31,32]. There is greater spending for subspecialist and home healthcare than for care provided by general pediatricians, who see the majority of these infants as outpatients. As expected, and previously shown [9], infants with chronic diagnoses of CLD and NEC/SIP experience a high burden of frequent follow-up and spending. These data may inform the counseling of families with medically complex neonates before discharge to help establish expectations and screening for needs that may facilitate timely early follow-up. Particularly for those infants with chronic conditions, financial burdens do not end at discharge. Families attend frequent follow-up appointments and accrue costs from travel, parking, and child care for other siblings [33], all of which can be continued stressors for families. Beyond the financial burdens of taking care of former NICU infants are the psychosocial burdens of high-volume interactions with the healthcare system and the impact on quality of life [34]; such aspects of NICU follow-up warrant additional exploration. Despite the higher number of visits than recommended when compared to a general population [19], the median cost per visit for an outpatient pediatrician encounter was similar to previously reported general data [35]. 

This study has several strengths. Notably, it addresses understudied, unanswered questions on how infants utilize early outpatient services following NICU discharge. Reassuringly, this work shows that recommendations for early follow-up are being followed by many. Our work may inform clinical recommendations and decision making around outpatient follow-up. It also provides evidence that supports robust counseling and anticipatory guidance for high utilizers of outpatient encounters such as infants with CLD and NEC. Finally, this study uses a large, national sample which captures a diverse group of infants, not just those who are preterm or who have medical complexities. About a third of infants were born at term in our study; they deserve representation in follow-up studies. 

Our work is not without limitations. This is a secondary data analysis, limited by claims’ encounter information and dependent on ICD codes for gestational ages and diagnoses; it thus lacks granular details on exact gestational ages, quality of encounters and definitive outpatient diagnoses. We reported outpatient encounters, which included a wide range of types of services, from primary care visits to lab encounters and supply center charges, which vary in their importance to the health and support of infants. The data examined included only the first three months of care after NICU discharge, which misses encounters, particularly those with subspecialists, that occur after this timeframe. Additionally, we were unable to differentiate between planned and unplanned readmissions; for example, we could not distinguish between planned surgeries versus unplanned illnesses or medical issues. Important social determinants of health were not available; these factors play a critical role in determining timely and consistent follow-up care [22,36]. Public insurance is a known risk factor for high use of medical services following NICU discharge [14] as well as lower adherence to follow-up recommendations [37]. Such evidence supports exploring early outpatient follow-up among infants with public insurance as this knowledge may be critical in providing more thorough support during the transition to the home. 

These findings stimulate additional questions. Descriptions of outpatient encounters which remain underexplored, such as home healthcare agencies/services, would be of interest. Outpatient follow-up characteristics among additional variables, including specific gestational ages, maternal demographics, and hospital characteristics, could help inform counseling and identifying at-risk groups. Prospective trials following infants from discharge to follow-up care would shed further light on follow-up patterns and challenges faced by families and medical home providers. Qualitative work from the parent’s perspective could also address challenges to initial follow-up care and transition to the home as well as the quality of follow-up care received, and provide insight into how to address families’ needs. In addition to readmission, other important NICU health outcomes should be studied as they relate to outpatient follow-up, such as adherence to immunization schedules, the monitoring of appropriate growth, breastfeeding rates, and the achievement of developmental milestones for both preterm and full-term infants. 

## 5. Conclusions

This large, national, retrospective study characterizes how former NICU infants utilize outpatient healthcare soon after discharge during a critical transition for patients and families from an intensive care unit to an outpatient setting. Our work shows that a majority of infants are seen within a couple days following discharge, with 90% seen within five days. The majority of early encounters are with a physician, primarily a pediatrician. The results are encouraging as they suggest that follow-up for many high-risk neonates is both timely and frequent. However, this places a burden on both families and the healthcare system, contributing to costly care. Further work can be conducted to ensure better adherence to follow-up recommendations for all former NICU patients.

## Figures and Tables

**Figure 1 children-11-00550-f001:**
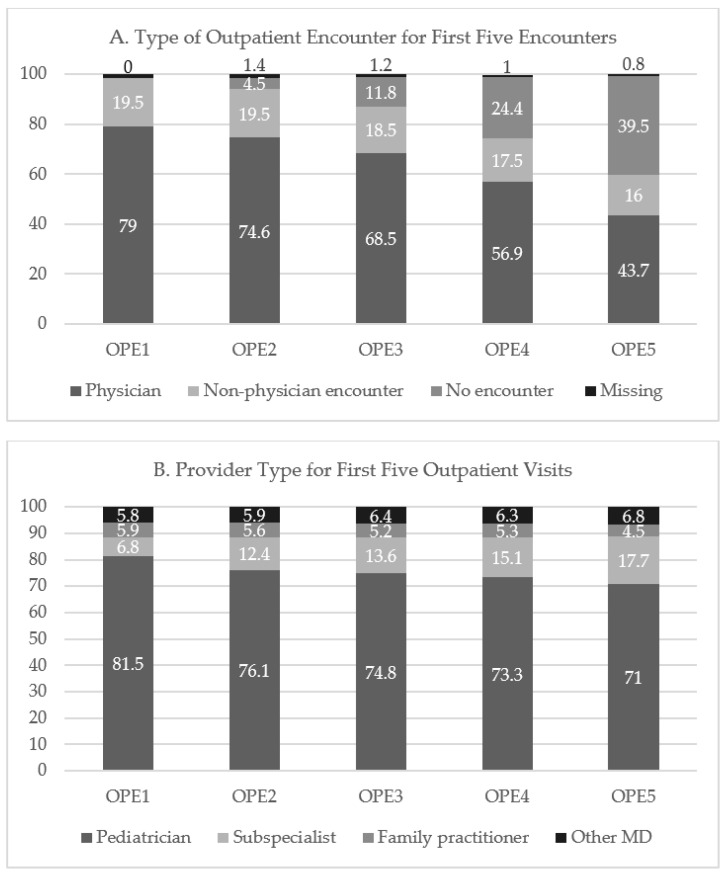
Type of outpatient encounter and provider type for first five encounters. (**A**) Type of outpatient encounter for first five encounters. Percentages of physician encounters (PEs), other encounters (OEs), no encounter, and missing data over the course of the first five OPEs. OEs include radiology, pathology, lab medicine, therapist (physical or other), pharmacy, supply center, home healthcare, and hospice. (**B**) Provider type for first five outpatient visits. Pediatrician encounters include pediatricians and mid-level providers. Subspecialists include both pediatric and non-pediatric subspecialists. Other MDs include internal medicine, multispecialty physician group, and unspecified medical doctors (Appendix A). Percentage of pediatric visits declined over time, while subspecialist visits increased. Family practitioner visits remained steady.

**Table 1 children-11-00550-t001:** Demographic and clinical characteristics of NICU patients.

	Frequency (%) or Median (Quartiles) and Range
Total N (%)	22,214
Gestational age Full-term Preterm	8058 (36.3) 14,156 (63.7)
Sex Male Female	12,759 (57.4) 9455 (42.6)
Multiples	3153 (14.2)
SGA/IUGR	648 (2.9)
NICU LOS (days)	
All admissions	8 (4, 21)
	1–269
Preterm	14 (7, 32)
	1–269
Full-term	4 (3–6)
	1–255
Year of birth 2015 2016 2017	8155 (36.7) 7691 (34.6) 6368 (28.7)
NICU diagnoses Respiratory distress BPD/CLD Sepsis Any congenital anomaly CHD Hypoglycemia Hemolytic disease HIE NEC/SIP IVH/PVL	7258 (32.7) 743 (3.3) 5610 (25.3) 6918 (31.1) 2839 (12.8) 3920 (17.6) 914 (4.1) 136 (0.6) 136 (0.6) 1666 (7.5)
Discharged with equipment ^1^	297 (1.3)
Readmitted in the first 3 months Readmit by 7 days Readmit 8–30 days	1745 (7.9) 693 (3.1) 558 (2.5)

^1^ GT, trach, oxygen, oxygen and ventilation, VPS. SGA/IUGR: small for gestational age/in utero growth restriction; LOS: length of stay; BPD/CLD: bronchopulmonary dysplasia/chronic lung disease; CHD: congenital heart disease; HIE: hypoxic–ischemic encephalopathy; NEC/SIP: necrotizing enterocolitis/spontaneous intestinal perforation; IVH/PVL: intraventricular hemorrhage/periventricular leukomalacia.

**Table 2 children-11-00550-t002:** Descriptors of outpatient encounters in the first three months following discharge.

	Frequency (%) or Median (Quartiles) and Range
Region of outpatient services Northeast Northcentral South West Unknown	6303 (28.4) 4853 (21.8) 8221 (37.0) 2735 (12.3) 102 (0.5)
Day of initial outpatient encounter	2 (1, 3)
	0–90
Day of first physician visit	2 (1, 3)
	0–90
Total number of encounters in first three months	143,743
All infants	5 (4, 7)
	1–90
Full-term	5 (3, 6)
	1–90
Preterm	5 (4, 8)
	1–90
Number of encounters in first three months by chronic NICU diagnosis ^1^	
CLD	10 (5, 17)
	1–87
CHD	6 (4, 10)
	1–90
NEC/SIP	8 (4,17)
	1–82
Brain injury ^2^	7 (4,11)
	1–88
Region of outpatient services Northeast Northcentral South West Unknown	6303 (28.4) 4853 (21.8) 8221 (37.0) 2735 (12.3) 102 (0.5)

^1^ Categories are not mutually exclusive; ^2^ brain injury includes HIE, IVH, PVL.

**Table 3 children-11-00550-t003:** Descriptors of patients with delayed care after NICU discharge.

Gestational Age at Birth	Preterm			Full-Term		
Time Outpatient Care Started	Delayed Care ^1^ N = 2653 (18.7)	Timely Care ^2^ N = 11,503 (81.3)	*p* Value	Delayed Care N = 1578 (19.6)	Timely Care N = 6480 (80.4)	*p* Value
**Frequency (%) or Median (Quartiles) and Range**						
Year of birth						
2015	1000 (37.7)	3938 (34.2)	<0.01	697 (44.2)	2520 (38.9)	<0.01
2016	975 (36.8)	4164 (36.2)		497 (31.5)	2055 (31.7)	
2017	678 (25.6)	3401(29.6)		384 (24.3)	1905 (29.4)	
Sex			0.52			0.93
Male	1488 (56.1)	6532 (56.8)		930 (58.9)	3809 (58.8)	
Female	1165 (43.9)	4971 (43.2)		648 (41.1)	2671 (41.2)	
NICU diagnoses						
CLD/BPD	85 (3.2)	644 (5.6)	<0.001	2 (0.1)	12 (0.2)	1.00
CHD	342 (12.9)	1590 (13.8)	0.22	175 (11.1)	732 (11.3)	0.86
IVH/PVL	295 (11.1)	1346 (11.7)	0.41	3 (0.2)	22 (0.3)	0.45
HIE	13 (0.5)	48 (0.4)	0.62	17 (1.1)	58 (0.9)	0.47
NEC/SIP	24 (0.9)	103 (0.9)	0.92	1 (0.1)	8 (0.1)	1.00
Congenital anomaly	808 (30.5)	3788 (32.9)	0.02	428 (27.1)	1894 (29.2)	0.10
Region of outpatient services						
Northeast	472 (17.8)	2941 (25.6)	<0.01	521 (33.0)	2369 (36.6)	0.01
Northcentral	584 (22.0)	2611 (22.7)		313 (19.8)	1345 (20.8)	
South	1261(47.5)	4451 (38.7)		527 (33.4)	1982 (30.6)	
West	332 (12.5)	1446 (12.6)		211 (13.4)	746 (11.5)	
Unknown	4 (0.2)	54 (0.5)		6 (0.4)	38 (0.6)	
Readmitted within three months	171 (6.4)	857 (7.5)	0.07	112 (7.1)	605 (9.3)	0.01
LOS for the first readmission, days	5 (2, 12)	4 (2, 10)	NS	3 (1.5, 7)	2 (1, 4)	NS
	1–205	1–349		1–87	1–174	

^1^ Delayed care: outpatient care started 4 or more days after NICU discharge; ^2^ timely care: outpatient care started on days 0–3 after NICU discharge.

**Table 4 children-11-00550-t004:** Predictors of readmission for term infants by three months following NICU discharge.

	Unadjusted				Adjusted			
Predictor	Odds Ratio	95% CI		*p* Value	Odds Ratio	95% CI		*p* Value
Congenital anomaly	2.71	2.32	3.17	<0.01	2.79	2.32	3.45	<0.01
Sex Female Male	Reference 1.01	0.87	1.18	0.89	0.97	0.83	1.13	0.69
SGA	0.89	0.52	1.52	0.67	0.87	0.51	1.49	0.61
Multiples	0.65	0.35	1.21	0.15	0.64	0.34	1.18	0.15
BI	0.54	0.22	1.34	0.14	0.51	0.20	1.27	0.15
CHD	2.00	1.63	2.44	<0.01	0.94	0.75	1.193	0.63
NICU LOS (days)	1.01	1.001	1.015	<0.03	1.00	0.99	1.01	0.77
Delayed care	0.74	0.60	0.92	<0.01	0.75	0.61	0.93	<0.01
Birth year 2015 2016 2017	Reference 0.89 0.93	0.74 0.77	1.07 1.13	0.22 0.47	0.91 0.94	0.76 0.78	1.10 1.14	0.32 0.52
Region Northeast Northcentral South West Unknown	Reference 1.31 0.92 1.27 0.00	1.07 0.76 0.99 0.00	1.61 1.12 1.62 0.00	0.01 0.42 0.06 1.00	1.27 0.88 1.20 0.00	1.03 0.72 0.94 0.00	1.56 1.07 1.54 0.00	0.02 0.20 0.15 1.00
Discharged with equipment ^1^	1.00	0.61	1.64	0.99	0.77	0.46	1.27	0.31

Hosmer–Lemeshow goodness of fit = 0.176. ^1^ GT, trach, oxygen, oxygen and ventilation, VPS.

**Table 5 children-11-00550-t005:** Predictors of readmission for preterm infants by three months following NICU discharge.

	Unadjusted				Adjusted			
Predictor	Odds Ratio	95% CI		*p* Value	Odds Ratio	95% CI		*p* Value
Congenital anomaly	3.26	2.86	3.71	<0.01	3.21	2.76	3.74	<0.01
NEC/SIP	3.02	1.93	4.74	<0.01	2.06	1.29	3.27	<0.01
SGA	0.45	0.27	0.75	<0.01	0.44	0.26	0.72	<0.01
Multiples	0.93	0.80	1.09	0.39	0.95	0.81	1.11	0.51
Sex Female Male	Reference 0.91	0.80	1.03	0.14	1.06	0.93	1.20	0.42
CLD	2.060	1.647	2.577	<0.01	1.200	0.93	1.55	0.16
CHD	2.033	1.744	2.371	<0.01	0.88	0.74	1.05	0.15
BI	1.334	1.115	1.595	<0.01	1.03	0.85	1.25	0.78
NICU LOS (days)	1.008	1.006	1.010	<0.01	1.003	1.000	1.005	0.02
Delayed care	0.86	0.72	1.014	0.07	0.90	0.76	1.070	0.23
Birth year 2015 2016 2017	Reference 1.05 0.90	0.91 0.77	1.22 1.06	0.52 0.22	1.07 0.94	0.92 0.79	1.25 1.11	0.38 0.47
Region Northeast Northcentral South West Unknown	Reference 1.07 0.91 1.11 0.94	0.90 0.77 0.90 0.34	1.29 1.08 1.38 2.62	0.44 0.28 0.33 0.91	1.034 0.85 0.98 0.92	0.86 0.72 0.78 0.33	1.25 1.01 1.22 2.610	0.73 0.06 0.83 0.88
Discharged with equipment ^1^	1.00	0.61	1.64	0.99	0.77	0.46	1.27	0.31

Hosmer–Lemeshow goodness of fit = 0.401. ^1^ GT, trach, oxygen, oxygen and ventilation, VPS.

**Table 6 children-11-00550-t006:** Spending on outpatient encounters in the first three months following NICU discharge.

A. Spending on first outpatient encounters							
	**All NICU Infants Regardless of Provider or Service** **(N = 22,214)**		**Peds** **N = 14,296**	**Subspecialty** **N = 1184**	**FP** **N = 1032**	**Other MD** **N = 1021**	**HHC** **N = 535**
Total Median, Quartiles	USD 7,234,881 USD 138 (103, 198)		USD 130 (102, 174)	USD 159 (101, 295)	USD 138 (109, 183)	USD 149 (107, 220)	USD 172 (120, 320)
B. Spending on outpatient services in first three months							
	**All Infants**	**Preterm**	**Full-Term**	**CLD**	**CHD**	**NEC/SIP**	**BI**
Total Median, Quartiles	USD 58,421,002 USD 1282 (831, 2383)	USD 1427 (876, 2711)	USD 1117 (772, 1855)	USD 3507 (1636, 7447)	USD 1953 (1029, 4245)	USD 3352 (1450, 7375)	USD 2065 (1047, 4471)

## Data Availability

The datasets presented in this article are not readily available because data were purchased from MarketScan Research Databases. Requests to access the datasets should be directed to them.

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
