# Peer review of "The Utilization of Early Outpatient Care for Infants Following NICU Discharge among a National Sample"

_children, 2024, doi:10.3390/children11050550_

Round 1

Reviewer 1 Report

Comments and Suggestions for Authors

I have read this paper with great interest, and value the analysis. I do have however some reflections and comments that are likely useful to further improve the messages of the paper.

First, as a non-US based reader, I would suggest to make it clearer (also in the title and the abstract), that this is a ‘national’ study, so likely representing a diversity of backgrounds, be it commercial insurance as common. This can be understood from the results and the discussion part of the paper.

I would like to somewhat better understand the framework of transition of care, as outpatient encounters should somewhat be initiated before discharge. This is somewhat mentioned in the paper, but is there a ‘structured’ approach on this, and how are outpatient care providers informed on the discharge and the medical/nursing aspects of the patients.

It is not clear to me if the database has any information on planned versus unplanned/emergency care ?

Assessment of activities based on billing information likely provided a clear snapshot on the presence of activities. However, it does not provide any information on the quality of care, or how the transition is organized. I have read that the authors somewhat reflect on this, with a call for additional studies. I would suggest to add qualitative type of studies to such a design, trying to understand the need of parents and care providers.

Editing

The abbr mentioned in tables should likely be explained in the table (subheading or legend)

Line 234: the greatest burden is perhaps not the best word choice, as parents also have their issues, while I assume that the majority of pediatricians are happy to assist ?

Author Response

I have read this paper with great interest, and value the analysis. I do have however some reflections and comments that are likely useful to further improve the messages of the paper.

First, as a non-US based reader, I would suggest to make it clearer (also in the title and the abstract), that this is a ‘national’ study, so likely representing a diversity of backgrounds, be it commercial insurance as common. This can be understood from the results and the discussion part of the paper.

“National sample” was added to the title, and last paragraph of the Introduction. Indication of a national study is present in the abstract.

I would like to somewhat better understand the framework of transition of care, as outpatient encounters should somewhat be initiated before discharge. This is somewhat mentioned in the paper, but is there a ‘structured’ approach on this, and how are outpatient care providers informed on the discharge and the medical/nursing aspects of the patients.

Good observations; recommendations on the transfer of care do exist and were added in the Discussion: “Recommendations exist on how to optimize the safe and streamlined transfer of care from the NICU to the outpatient providers and include identifying a primary care provider, often a pediatrician, communicating pertinent medical information through written or oral means, and following up with families after discharge [1].”

It is not clear to me if the database has any information on planned versus unplanned/emergency care ?

This is correct – we are not able to distinguish between planned and unplanned care and this was added as a limitation with the line in the Discussion: “Additionally, we are unable to differentiate between planned and unplanned readmissions; for example, we cannot distinguish between planned surgeries versus unplanned illnesses or medical issues.”

Assessment of activities based on billing information likely provided a clear snapshot on the presence of activities. However, it does not provide any information on the quality of care, or how the transition is organized. I have read that the authors somewhat reflect on this, with a call for additional studies. I would suggest to add qualitative type of studies to such a design, trying to understand the need of parents and care providers.

This is an excellent idea and methodology to achieve such perspective. The following line was added to the last paragraph of the Discussion on future directions: “Qualitative work from parent’s perspective could also address challenges to initial follow up care and transition home, quality of follow up care received, and insight into how to address families’ needs.”

Editing

The abbr mentioned in tables should likely be explained in the table (subheading or legend)

Abbreviations were added to the table subheading

Line 234: the greatest burden is perhaps not the best word choice, as parents also have their issues, while I assume that the majority of pediatricians are happy to assist ?

Good point – the word “burden” was changed to “responsibility”

Reviewer 2 Report

Comments and Suggestions for Authors

Dear Authors

An important research that needs some more data to improve. 

State the level of prematurity in each case. Do all cases follow the same follow-up?

The purpose of your study is not clear and the references of the introduction are quite old. 

The methodology also divides infants into full-term and preterm. But isn't this division misleading? There are 3 categories of prematurity and you have to take into account the reason for NICU admission and complications.

The table of demographics could be obtained and the data discussed

In conclusion, additional research is needed that will study additional variables such as the socio-economic status of the family, the degree of prematurity, co-morbidity, and hospital

Author Response

Dear Authors

An important research that needs some more data to improve. 

State the level of prematurity in each case. Do all cases follow the same follow-up?

The data does not specify the gestational ages of all neonates; thus, we categorized as preterm versus term. In regards to follow up, recommendations are not specific to term versus preterm. This was clarified in the manuscript in the second paragraph of the Introduction with the line: “The American Academy of Pediatrics recommends that, prior to discharge, appropriate follow up be established with primary care physicians, subspecialties, and skilled home nursing care, and that other necessary coordination occur for needed medication and home equipment, without specifying recommendations based on gestational age or diagnoses.”

The purpose of your study is not clear and the references of the introduction are quite old. 

The following line was edited in the Abstract: “The objective of this secondary data analysis is to describe outpatient encounters (OPE) within the first three months following discharge of commercially insured infants admitted to NICUs in the MarketScan Research Database nationally from 2015-2017.” The following line was edited in the Introduction: “In this work our objectives were to describe aspects of early outpatient utilization within three months of NICU discharge among a national sample including timing of initial encounter, frequency of encounters, location of services, association of timing with early readmission, and spending on outpatient encounters.”

The following references were added/adjusted:

  • Reference #18 updated to from 1998 to 2008
  • Reference #4 updated to manuscript on readmissions among very low birth weight from 2019
  • Reference #5 updated to cost paper on preterm neonates from 2016

The methodology also divides infants into full-term and preterm. But isn't this division misleading? There are 3 categories of prematurity and you have to take into account the reason for NICU admission and complications.

Reasons for NICU admission and complications do vary among term and preterm neonates, which was in part the reason for breaking down the regression models into term and preterm, such as NEC among preterms being higher than in term neonates. Exact gestational ages were not available in the database for all patients and were thus unable to be included in analyses. Term and preterm was reported. 

The table of demographics could be obtained and the data discussed

Additional descriptors of Table 1 were added in the results, including percentages of infants’ birth years and frequency of other clinical diagnoses.

In conclusion, additional research is needed that will study additional variables such as the socio-economic status of the family, the degree of prematurity, co-morbidity, and hospital

Excellent points. The following statement was added to the future directions paragraph in the Discussion: “Outpatient follow up characteristics among additional variables including specific gestational ages, maternal demographics, and hospital characteristics would be of interest.”

Round 2

Reviewer 2 Report

Comments and Suggestions for Authors

Dear Aurhors 

You did a great job! Congratulations!

Comments on the Quality of English Language

Minor